# Potential Plasma Proteins (LGALS9, LAMP3, PRSS8 and AGRN) as Predictors of Hospitalisation Risk in COVID-19 Patients

**DOI:** 10.3390/biom14091163

**Published:** 2024-09-17

**Authors:** Thomas McLarnon, Darren McDaid, Seodhna M. Lynch, Eamonn Cooper, Joseph McLaughlin, Victoria E. McGilligan, Steven Watterson, Priyank Shukla, Shu-Dong Zhang, Magda Bucholc, Andrew English, Aaron Peace, Maurice O’Kane, Martin Kelly, Manav Bhavsar, Elaine K. Murray, David S. Gibson, Colum P. Walsh, Anthony J. Bjourson, Taranjit Singh Rai

**Affiliations:** 1Personalised Medicine Centre, C-TRIC Building, Altnagelvin Area Hospital, School of Medicine, Ulster University, Glenshane Road, Derry-Londonderry BT47 6SB, UK; 2School of Computing, Engineering & Intelligent Systems, Ulster University, Derry BT48 7JL, UK; 3School of Health and Life Sciences, Teesside University, Campus Heart, Middlesbrough TS1 3BX, UK; 4Altnagelvin Area Hospital, Western Health and Social Care Trust, Derry BT47 6SB, UK; 5Clinical Chemistry Laboratory, Altnagelvin Hospital, Derry BT47 6SB, UK; 6Biomedical Sciences Research Institute, University of Ulster, Coleraine BT52 1SA, UK

**Keywords:** SARS-CoV-2, COVID-19, biomarker, LGALS9, LAMP3, PRSS8, AGRN, Support Vector Machine, Logistic Regression, Random Forest

## Abstract

*Background:* The COVID-19 pandemic, caused by the novel coronavirus SARS-CoV-2, has posed unprecedented challenges to healthcare systems worldwide. Here, we have identified proteomic and genetic signatures for improved prognosis which is vital for COVID-19 research. *Methods:* We investigated the proteomic and genomic profile of COVID-19-positive patients (n = 400 for proteomics, n = 483 for genomics), focusing on differential regulation between hospitalised and non-hospitalised COVID-19 patients. Signatures had their predictive capabilities tested using independent machine learning models such as Support Vector Machine (SVM), Random Forest (RF) and Logistic Regression (LR). *Results:* This study has identified 224 differentially expressed proteins involved in various inflammatory and immunological pathways in hospitalised COVID-19 patients compared to non-hospitalised COVID-19 patients. LGALS9 (*p*-value < 0.001), LAMP3 (*p*-value < 0.001), PRSS8 (*p*-value < 0.001) and AGRN (*p*-value < 0.001) were identified as the most statistically significant proteins. Several hundred rsIDs were queried across the top 10 significant signatures, identifying three significant SNPs on the *FSTL3* gene showing a correlation with hospitalisation status. *Conclusions:* Our study has not only identified key signatures of COVID-19 patients with worsened health but has also demonstrated their predictive capabilities as potential biomarkers, which suggests a staple role in the worsened health effects caused by COVID-19.

## 1. Introduction

COVID-19 has placed constraints on many healthcare systems with over 770 million documented cases since its onset in 2019 [1]. COVID-19 is caused by the SARS-CoV-2 virus when its spike protein binds to the ACE2 receptor, allowing access into the cell and beginning infection [2]. Upon infection onset, viral Pathogen-Associated Molecular Patterns (PAMPs) are detected by specific endosomal pattern recognition receptors such as Toll-like receptors, resulting in intracellular signalling responses which activate various inflammatory signatures such as NF-kB, IRF and IFN, causing extreme cytokine promotion [2,3]. These collective inflammatory signatures are thought to be indicators of disease severity, which has been referred to as the cytokine storm [4]. Symptoms such as fever, cough, fatigue, shortness of breath, sore throats and headaches are common amongst COVID-19-positive patients, and other adverse symptoms include gastrointestinal problems, acute respiratory distress syndrome, septic shock, and coagulation dysfunction, which are dependent on severity and co-morbidities [5].

Recent studies show the impact clinical differences such as age, weight, specific comorbidities, and socioeconomic deprivation play as associated risk factors for hospitalisation in COVID-19 patients [6]. Patients with varying severities, clinical differences and differing symptomologies demonstrate the importance of identifying specific biological signatures that are associated with disease severity. Identifying the biological factors that govern these alternative outcomes is essential for risk assessment, tailored treatment and pandemic control. Biological markers such as LDH, D-Dimer and IL-6, which are significantly differentially expressed in hospitalised COVID-19 patients, are currently used in clinical care [7]. There is, however, an absence of information on a wider range of biological signatures that are not currently employed in clinical care. Expanding the list of signatures investigated allows for more potential to discover biological markers that could perform equal to or better than currently used markers, which would improve patient management, prognosis and allocation of resources, respectively.

Data used throughout this investigation was collected from patients participating in the COVID-19 Response Study, wherein hospitalised and non-hospitalised COVID-19 patients with varying clinical levels of severity were recruited and proteomic differences were analysed [8]. Proteomic differences were quantified using OLINK plasma proteomics, which provides a minimally invasive means to assess systemic alterations during COVID-19 infection. Many existing proteomic studies have identified differentially expressed signatures in severe COVID-19 patients across different cohorts. Whole-Genome Sequencing (WGS) was performed on patients to identify potential SNPs, which are correlative to hospitalisation risk. We analysed the rsIDs of the top 10 significant signatures identified from the differential regulation analysis and searched several hundred SNPs across these signatures.

We recruited 500 COVID-19-positive patients from the COVID-19 Response Study [8] (n = 400 proteomics) and quantified their plasma proteins through OLINK proteomics using the OLINK Explore 384 Inflammatory panels between hospitalised and non-hospitalised cohorts. Patients selected for omics analysis were both age- and sex-matched, which allowed for biological differences to be accounted for. Identified proteins had their predictive capabilities measured both individually and collectively using various supervised machine learning approaches. For WGS (n = 483), patients had saliva samples taken and DNA isolated which was then used in the building of libraries for genome analysis. Very few studies have combined biomarker discovery with machine learning methodologies. Here, we applied multiple machine learning methods to predict the most significant biomarker combinations that can stratify patients based on hospitalisation risk.

## 2. Materials and Methods

### 2.1. Patient Recruitment

Five hundred patients were recruited to the COVID Response Study (COVRES): A Northern Ireland population study of SARS-CoV-2 prevalence, predisposing factors and pathology (approved by the Health and Care Research Wales Ethics service; REC ref 20/WA/0179). Written informed consent was taken from all patients. Patient material and information usage were approved by ORECNI (Project Reference Number: IRAS 283596). The anonymised patient data and corresponding clinical data were collated for all patients recruited as part of COVRES, with this investigation using only patients (n = 400) who had their plasma protein samples measured [8].

### 2.2. Handling of Biological Samples

#### 2.2.1. OLINK Plasma Protein Quantification

From the 500 patients recruited, patients were age- and sex-matched; 400 patient samples were selected for plasma protein analysis. Blood samples were collected in triplicate in 10 mL EDTA tubes. Samples were fractionated by centrifuging at 4000 rpm at 4 °C for 15 min, with the plasma being isolated and aliquoted into cryovials (stored at −80 °C) for later analysis. Plasma samples were thawed at room temperature (20 °C) and 45 µL of each sample were pipetted into 96-well plates with 8 wells left empty on each individual plate for OLINK’s own controls which consist of negative controls, incubation controls and inter-plate controls. Once samples were plated, they were virus-inactivated following OLINK COVID-19 1% TritonX-100 inactivation protocol. All plates were sealed using an adhesive sealer. Plates were vortexed thoroughly before centrifugation at 400× *g* for 1 min at 20 °C prior to a 2-h incubation period at room temperature (20 °C). Once incubation was finished, plates were stored at −80 °C until shipment. Upon receiving samples, OLINK added their inter-controls to each plate and proteomic quantification of 400 plasma samples was performed using the Explore^®^ 384 inflammation panel (OLINK, Boston, MA, USA). Overlapping assays of IL-6, IL-8 (CXCL8) and TNF are included for quality control (QC) purposes at OLINK. Due to OLINK’s in-house normalisation protocol, all units of measurement for the plasma proteins are Normalised Protein Expression (NPX), which is at log2 scale.

#### 2.2.2. DNA Isolation

DNA from saliva was isolated using PrepIT L2P (DNA Genotek, Ottawa, ON, Canada). Extracted DNA was quantified using the Qubit 3.0 fluorometer (Thermo Scientific, Boston, MA, USA) and the NanoDrop 1000 spectrophotometer (Thermo Scientific, Boston, MA, USA) with sequencing conducted using the Invitrogen Quant-iT PicoGreen dsDNA Assay Kit (Invitrogen, Paisley, UK) (P7589) on the Hamilton Microlab Star before storage at −80 °C.

### 2.3. Genome Analysis

Whole-genome library preparation was performed using the Illumina TruSeq PCR Free Library Prep protocol (20015963) using a sample volume input of 1 µg (n = 483 passed QC). Library preparation was automated and processed using a Hamilton NGS Star robotic workstation (Hamilton company, Birmingham, UK). Library quality was assessed using the Roche KAPA Library Quantification Kit (Roche, Hertfordshire, UK) (7960298001). Libraries were pooled and sequenced (150 bp paired end (PE)) on an Illumina NovaSeq 6000 instrument using the NovaSeq 6000 S4 Reagent Kit (Illumina, Cambridge, UK) v1.5 (20028312), targeting a mean coverage of 30x The pipeline for data preparation is illustrated in the graphical abstract. Raw data (BCL format) were demultiplexed and converted to FASTQ format using BCL2FastQ (Illumina, Cambridge, UK). Adapters were trimmed using Skewer (v0.2.2) and quality control was assessed using FASTQC. Data was down-sampled by selecting the first 800 million reads per sample when necessary. Secondary analysis was performed using an automated Sention (GoldenHelix Ò)-based pipeline. FASTQ data were aligned to the GRCh38/hg38 reference genome build via the Burrows-Wheeler Aligner (BWA). Alignment quality was assessed, reads sorted, duplicates marked, indels realigned and base quality scores recalibrated. Sequences were uploaded onto the European Genome-phenome Archive (EGA). Data was stored according to genomic position in the Genuity Science Genomically Ordered Relational database (GORdb) to facilitate rapid access by the Clinical Sequence Analyzer™ user interface and Sequence Miner visualisation software (v11.0.0), which was used for variant analysis. Further, dbSNP was used to identify SNPs of interest on our pre-identified signatures. Then, a targeted approach was used for extracting patient data for the 10 identified SNP variants using Linux Bcftools and Bgzip, which enabled us to carry out variant calling and SNP extraction.

Subsequently, patient IDs were mapped to phenotypic data (non-hospitalised and hospitalised).

Then, a targeted approach was used for extracting patient data for the 10 identified SNP variants using Linux Bcftools (v1.14) and Bgzip (v1.14), which enabled us to carry out variant calling and SNP extraction.

### 2.4. Statistical Testing

We investigated hospitalised vs. non-hospitalised patients and can logically deduce that the proteomic regulation differs between these groups due to differing inflammatory responses, resulting in unequal variances and requiring the use of Welch’s two-tailed *t*-test. We tested all proteins on the inflammatory panel obtaining respective log2FC, *p*-values, t-values and degrees of freedom for each. Calculating log2FC on the NPX data was done using as follows; mean hospitalised cohort − mean non hospitalised, as data was already log2-transformed, logging the mean differences wasn’t required. False discovery rates (FDRs) were calculated using Benjamini-Hochberg’s correction on the identified *p*-values.

### 2.5. Differential Regulation Analysis

We first visualised the differences in proteomic regulation using a volcano plot with the −log10 (*p*-value) plotted on the *y*-axis and the log2FC on the *x*-axis. This was done to visualise the overall proteomic regulation changes between the hospitalised and non-hospitalised cohorts. Heatmaps were then employed, consisting of patients ordered appropriately according to hospitalisation status and protein, which were then clustered as rows. This allowed us to visualise the individual patients’ protein changes across hospitalisation status. NPX values were scaled using Z-score normalisation, ensuring all proteins were on the same scale individually. Violin box plots were then employed to gain better visualisation of the overall and individual protein regulation changes between hospitalisation status for the most statistically significant proteins.

### 2.6. Proteomic Separation

Initially, we used Principal Component Analysis (PCA) to plot all patients (hospitalised vs. non-hospitalised) based on all their recorded proteomic measurements to see if they would separate. We then used proteomic regulation scatter plots of the most significant three proteins, with the *x-*, *y-* and *z*-axes all representing NPX of specific proteins labelled by hospitalisation status.

### 2.7. Pathway Analysis

Pathway analysis was performed using pathfindR (v2.4.1), a bioinformatics package based on R (v4.3.1) that conducts enrichment analysis through active subnetworks. Input for this analysis was gene symbol, log2FC and false discovery rate, which was identified by applying Benjamini-Hochberg’s correction method to the pre-existing *p*-values from the differential regulation analysis. The analysis yielded a pathway analysis plot that had fold-enrichment on the *x*-axis, each gene ontology term on the *y*-axis, and the size of each data point was dependent on the number of input genes found in that specific GO term.

### 2.8. Machine Learning

Testing the predictive capabilities of the proteins was the next step in identifying biomarkers of COVID-19 hospitalisation status. We employed three supervised machine learning models in the form of Support Vector Machine (SVM), Logistic Regression (LR) and Random Forest (RF) to predict hospitalisation risk univariately and then used feature selection tools such as Recursive Feature Elimination (RFE) and Least Absolute Shrinkage Selection Operator (LASSO) to see if predictions can be improved by using an optimal protein panel.

Each protein on the OLINK panel had its own SVM, LR and RF model built using only each protein’s NPX values. Initially, the dataset was split into training (80%) and testing (20%) cohorts, with the training data being used to fit each model. SVM had its kernel set to linear and had its cost parameter tuned; LR had its family set to binomial and RF had the number of trees tuned. These tuning methods were all carried out uniquely for each protein so the model could perform optimally using a single protein. Ten-fold cross-validation was conducted on each model for each protein, and then the model was set to predict the unseen test data. Confusion matrix tests were used to measure each model’s predictive accuracy, *p*-value, sensitivity and specificity when predicting test data, which provided statistics on each protein’s predictive capabilities across the three models. The predicted probability values for each protein across three models were then used alongside the unseen hospitalisation statuses to generate three Receiver Operator Characteristic (ROC) curves for each protein across the three models. Then, 95% confidence intervals were generated on the ROC curves and were constructed by bootstrapping the sensitivities and specificities of each protein’s predictions 500 times for each model. Proteins were then tested in combination using the above machine learning models with different feature selection tools applied. RFE was applied to both SVM and RF, while LASSO was applied to LR. The final models were then used to generate ROC curves and identify which the best proteins were. The code is fully public and can be accessed (link to code on github).

### 2.9. Protein Association Analysis

Potential associations between key proteins were identified through stringDB, a proteomic association database supported by evidence from known databases, experimentally proven interactions, gene neighbourhoods, gene co-occurrences, gene fusions, text mining, co-regulation and protein homology. Proteins identified as significant from the previous analysis were queried in stringDB, returning a protein interaction network that was expanded until the most significant proteins were interconnected.

## 3. Results

### 3.1. Demographics of COVID-19 Cohort

Differential proteomic analysis paired with the machine learning predictions carried out within the study are portrayed in the Graphical abstract as an overarching study design. Physiological differences between patients were accounted for by performing statistical tests on the base demographical information such as age, gender and severity across hospitalisation status (Table 1, column 4). Age and gender were identified as significant, with more males becoming hospitalised with COVID-19 (120) compared to females (90) (*p* < 0.05).

### 3.2. Differentially Expressed Proteins in Hospitalised Patients

Proteomic differences between hospitalised and non-hospitalised patients were performed using Welch’s two-tailed *t*-test on protein regulation against hospitalisation status. The WHO Severity score and biological age were also used when visualising proteomic differences with clinical overlap, which demonstrated a near-perfect overlap between the WHO classification of severity and hospitalisation status. This shows data collection accuracy was maintained throughout the study, as the WHO classification for severity corroborates with the hospitalisation of patients. The differential proteomic analysis yielded 30 highly significant proteins that were overexpressed in hospitalised COVID-19 patients compared to non-hospitalised patients, which passed a significance threshold of log2FC >= 0.5 and *p*-value < 0.01 (Table 2, columns 2 and 3, respectively). The top ten proteins identified included LGALS9, PRSS8, AGRN, LAMP3, TREM2, TNFRSF11A, LAIR1, FSTL3, FABP1 and HGF. These proteins were then visualised using heatmaps to identify the individual regulation of each significant protein across all patients, with the protein regulation undergoing z-score normalization to better visualise the magnitude of the regulation (Figure 1A). Next, we visualised the proteomic regulation of patients on a volcano plot to better identify which proteins had the largest change in regulation according to their log2FC and how significant each was according to their −log10(*p*-Value). The statistical threshold was log2FC >= 0.5 and *p*-value < 0.05, with 31 significant proteins identified. LGALS9, LAMP3, AGRN and PRSS8 were found to be the most significant (Figure 1B). Violin box plots were then employed to visualise these proteins, showing individual protein regulations of each patient, the results of which aligned with previous findings (Figure 1C).

### 3.3. COVID-19 Patient Separation and Differential Signalling

PCA was used to separate patients based on hospitalisation status using all proteomic recordings. We observed significant separation on the *y*-axis of the PCA that followed a trend of non-hospitalised patients being above hospitalised patients on the second principal component (Figure 2A). We then employed a proteomic scatter plot of the most significant proteins to see the relationship between each other and hospitalisation risk. First, we plotted LGALS9, which was the most statistically significant protein, against LAMP3, which was one of the most statistically significant but also had one of the largest log2FC values (Figure 2B). We found there is a linear relationship between these proteins and hospitalisation risk. To better understand this relationship, we added a third protein, PRSS8, which was another statistically significant protein with a large log2FC (Figure 2C). A positive correlation was observed between all three of these proteins with hospitalisation risk, demonstrating an observable correlation between these protein signatures and hospitalisation risk. We then queried LGALS9, LAMP3, PRSS8 and AGRN in stringDB to identify any potential interactions through the generation of a protein-protein interaction network (Figure 2D). When the protein network was expanded, LGALS9, LAMP3 and AGRN were found to be interconnected via CD44 and CTLA4, with evidence from text mining and co-regulation. Lastly, pathway analysis was then performed using pathfindR to better identify the changes in biological systems between hospitalised and non-hospitalised patients. As expected, viral protein interaction with cytokine and cytokine receptor, cytokine-cytokine receptor interaction and chemokine signalling pathway with the cytokine-cytokine receptor interaction pathway were enriched (Figure 2E).

### 3.4. Univariate Machine Learning Predictions for Hospitalisation Risk

After identifying LGALS9, PRSS8, LAMP3 and AGRN as being the most significant from our differential regulation analysis, we then wanted to test their predictive capabilities, both individually and collectively. We first built three independent, supervised machine learning models for each protein, wherein the only feature used to train the models was each protein’s NPX regulation values. Each protein had its predictive capabilities assessed by taking the predicted probability values from each model and generating ROC curves to test their AUC scores. Models consisted of Support Vector Machines, Logistic Regression and Random Forests. LGALS9 scored AUC scores of 0.82 for both SVM and LR, and 0.74 for RF; LAMP3 scored AUC scores of 0.82 for both SVM and LR, and 0.75 for RF; PRSS8 scored AUC scores of 0.80 for both SVM and LR, and 0.75 for RF; and AGRN scored AUC scores of 0.82 for both SVM and LR, and 0.69 for RF (Figure 3). The predicted probabilities for the top proteins differed between SVM and LR, but in spite of this, their generated AUC scores are identical and have near-identical confidence intervals. This suggests that the SVM and LR models effectively captured the linear relationship between the proteins and the hospitalisation status of patients. We acknowledged that the results for the univariate RF analysis were not as effective as SVM or LR and attributed this occurrence to the way RF primarily handles highly complex data; with only a single feature within each model, the decision trees generated can only make decisions based on that single feature, resulting in poorer classification performance.

### 3.5. Feature Selected Machine Learning Predictions for Hospitalisation Risk

After assessing the effectiveness of these proteins at predicting hospitalisation risk individually, we then wanted to test their combined predictive capabilities when they were feature-selected with other proteins. RFE was used for both SVM and RF, whereas LASSO was used for LR. The final models were investigated, with the top 10 proteins in each model being ranked according to variable importance score, of which LGALS9, LAMP3, PRSS8 and AGRN were present in the top 10 proteins used to build the model across SVM, LR and RF (Figure 4A). The final models then had ROC curves generated to assess their AUC scores as a metric for predictive capability (Figure 4B), with confusion matrices also being generated to test false positive rates and true positive rates (Figure 4C). The AUC scores for the models were within an acceptable range, with the RFE-SVM, RFE-RF and LASSO-LR models returning scores of 0.849, 0.835 and 0.856, respectively (Figure 4B). In comparison to the feature-selected models with the univariate models, the feature-selected models attained better performance metrics across all three compared to their univariate counterparts.

### 3.6. SNPs on Genes of Interest

We were interested in finding any SNPs on corresponding genes of these identified significantly altered proteins. The purpose of SNP identification was to gain insights into the mechanistic changes driving the severity of COVID-19. Finally, querying several hundred SNPs across our identified signatures only yielded one gene with significant hospitalisation correlation, *FSTL3*. Three SNPs were identified (rs1046253, rs2057713, rs2057714, *p*-values < 0.001) showing different hospitalisation statuses depending on the genotype. The rs1046253 SNP shows differing genotypes for clinical outcomes, with the heterozygous genotype being more prevalent in hospitalised patients compared to non-hospitalised patients and the homozygous alternative genotype being more prevalent in non-hospitalised patients compared to hospitalised patients (Figure 5A). The inverse is true for the remaining two SNPs (rs2057713/rs2057714) as the heterozygous genotype is more prevalent in non-hospitalised patients than hospitalised patients and the homozygous alternative genotype is more prevalent in hospitalised patients than non-hospitalised patients (Figure 5B,C).

## 4. Discussion

The severity of COVID-19 symptoms is dependent on the biological and physiological status of the infected patient, making it difficult to identify patients likely to experience worsening health outcomes. Our study shows several plasma proteins as potential predictors of health risk in the form of hospitalisation in COVID-19 patients. Our stricter differential proteomic analysis identified 30 significant plasma proteins with a log2FC >= 0.5 and a *p*-value < 0.01, with LGALS9, LAMP3, PRSS8 and AGRN all being upregulated in hospitalised patients compared to non-hospitalised patients. These proteins then had their individual predictive capabilities assessed across three independent machine learning models and attained similar validation metrics across each. The feature-selected models had improved AUC scores for predicting risk in the form of hospitalisation. When the proteins used as features were analysed within each model it was found that LGALS9, LAMP3, PRSS8 and AGRN were all found within the top 10 variables of greatest importance to each of the models. LGALS9 was identified as the most important protein across all three of the independent models, which suggests its potential as a biomarker, both individually (individual protein ML model) and within a panel of other proteins (feature selected protein ML model).

LGALS9 (Galectin-9) is a member of the galectin family and is a commonly expressed protein in immunological regulation [9]. Other studies have shown its involvement in COVID-19, with it being more promoted in severe COVID-19 patients [6,10]. Similarly, LAMP3, a lysosomal glycoprotein that is implicated in adaptive immunity [11], and PRSS8, a cleavage protein with a preference for poly-basic substrates [12], have been found to be associated with COVID-19 severity, as LAMP3 is considered a predictor of severe COVID-19 development, whereas PRSS8 was linked being highly expressed in extra-cellular vesicles of severe-COVID-19 patients [7,8]. While other studies demonstrate that regulation levels of LGALS9 are distinguishable between healthy controls and COVID-19-positive patients, we demonstrate that LGALS9 levels are not only associated with COVID-19 but also the severity of the disease according to patient hospitalisation.

Our results also coincide with these published findings and show that LGALS9, LAMP3 and PRSS8 are more expressed in hospitalised COVID-19 patients compared to non-hospitalised ones and demonstrated its utility as a predictive biomarker of health risk. We go on to show that AGRN is also differentially expressed in hospitalised COVID-19 patients and has good predictive capabilities. AGRN (Agrin) is a proteoglycan that is involved in the neuromuscular system, but more recent literature suggests it plays a role in the NOTCH signalling pathway, which could implicate it in immunological responses, as other literature suggests its role in immunological responses to acute COVID-19 in high-risk patients [13,14,15,16]. When querying key proteins to find known interactions on stringDB, we found that LGALS9, LAMP3 and AGRN are interconnected when the network is expanded to 14 proteins, suggesting a potential signalling mechanism that could be responsible for the inflammatory differences between hospitalised and non-hospitalised COVID-19 patients [17]. LGALS9 is interconnected with AGRN via CD44, a cell surface receptor that is involved in cell signalling and adhesion migration and has involvement in inflammation [18]. Studies also show that CD44 is highly expressed in CD4+ and CD8+ T cells of severe COVID-19 patients, which further supports the interconnected relationships these proteins have within the immunological response to COVID-19 [19]. LGALS9 is also connected to LAMP3 within the network via CTLA4, an inhibitory receptor that negatively regulates T-cell responses and is also connected to CD44 within the network [20].

After identifying key signatures within hospitalised COVID-19 patients, we then set out to discover if there are any SNPs responsible for the differing phenotypic events. We queried multiple SNPs across our top 10 differentially expressed signatures, but only *FSTL3* yielded significant SNPs. These SNPs were found to be significantly correlated with COVID-19 hospitalisation risk, with differing genotypes being more prevalent in distinct cohorts. *FSTL3* has been implicated as a marker of COVID-19—CVD complications [21]; however, no SNPs were found to be significantly correlated with hospitalisation for *LGALS9*, *LAMP3*, *PRSS8* or *AGRN*.

With our own findings coinciding with the current literature, LGALS9, LAMP3, PRSS8 and AGRN prove to be potential biomarkers of health risk in COVID-19 patients, both individually and collectively. With these plasma proteins having the potential to be biomarkers, it is important that further validation of these signatures is carried out. Future work from this project would consist of validating these proteins using different biological data with different machine learning techniques. Potential validation would entail RNA sequencing for transcriptomic quantification. Transcriptomic data would be used for supervised machine learning assessment, which would be utilised to test if proteomic and transcriptomic predictions align. Combining multiple omic datasets and testing their predictive capabilities across multiple machine learning models would ensure that overlapping signatures would attain notoriety as identified markers of health risk in COVID-19.

The integration of machine learning analysis with proteomic data enhances our understanding of COVID-19 pathogenesis and offers a promising avenue for personalised medicine. The identified proteins, when incorporated into predictive models, can aid clinicians in risk stratification and treatment decisions. For instance, individuals at higher risk of developing severe disease can be identified early in their illness, allowing for proactive interventions and resource allocation. Furthermore, the machine learning models provide an adaptable framework that can be continually refined as more data become available, ensuring their relevance and accuracy throughout the evolving course of the pandemic. This approach represents a significant step toward precision medicine in COVID-19, tailoring interventions to individual patient profiles.

## 5. Conclusions

In conclusion, our study demonstrates that LGALS9, LAMP3, PRSS8 and AGRN are highly significant, differentially regulated plasma proteins found in hospitalised COVID-19 patients. We have shown their effective predictive capabilities, both individually and collectively, across three distinct supervised machine learning models with two different feature selection tools. The results of these all coincided with each other, suggesting that these proteins are viable biomarkers of COVID-19 hospitalisation risk.

## Figures and Tables

**Figure 1 biomolecules-14-01163-f001:**
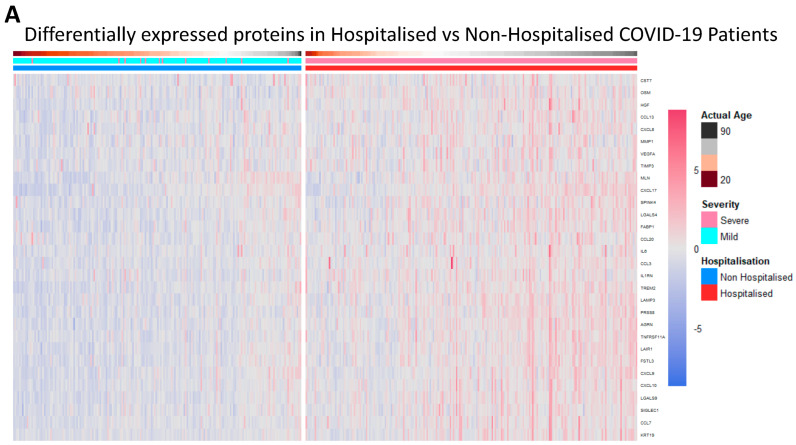
Differentially expressed proteins in hospitalised patients. (**A**). Heatmap with patients being grouped in columns according to their hospitalisation status, severity status according to the WHO scale (1–4 mild, 5–10 severe), and age. Proteins clustered as rows, with the significant threshold for proteins set to log2FC > 0.5 and a *p*-value < 0.01. (**B**). Volcano plot of differentially expressed proteins in hospitalised patients compared to non-hospitalised patients, ranked according to their −log10(*p*-Value) on the *y*-axis and log2FC on the *x*-axis. The significance threshold was set to log2FC > 0.5 and *p*-value < 0.05. (**C**). Violin box plots of LGLAS9, LAMP3, PRSS8 and AGRN, depicting NPX regulation between hospitalised and non-hospitalised patients.

**Figure 2 biomolecules-14-01163-f002:**
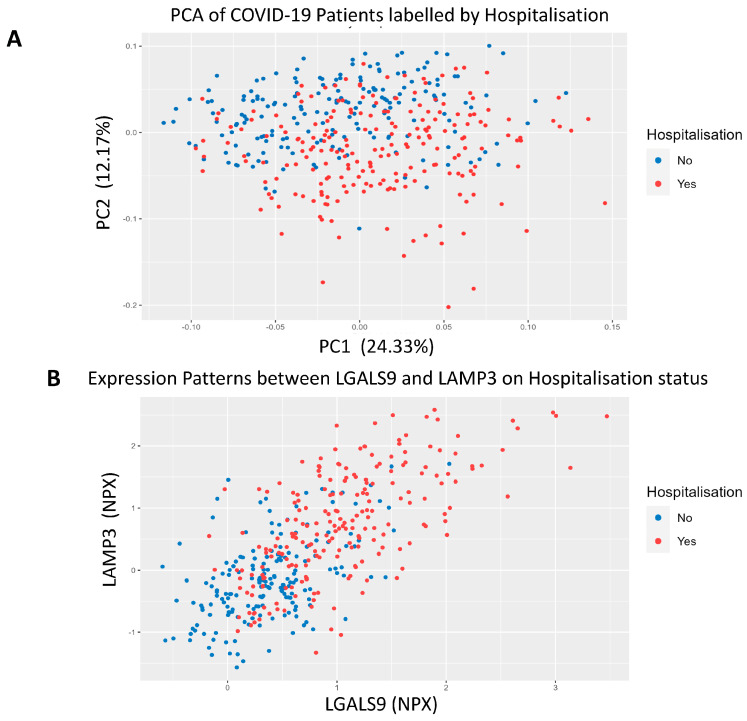
COVID-19 Separation and Signalling Differences. (**A**). Principal Component analysis of COVID-19 patients using all proteomic values. The *x*-axis represents PC1, which accounts for the most variance, and the *y*-axis represents PC2, which accounts for the second most variance, labelled according to hospitalisation status. (**B**). 2D-proteomic scatter plot depicting NPX regulation of LGALS9 on the *x*-axis and LAMP3 on the *y*-axis for each patient, labelled according to their hospitalisation status. (**C**). 3D-proteomic scatter plot depicting NPX regulation of LAMP-3 on the *x*-axis, LGALS9 on the *y*-axis and PRSS8 on the *z*-axis for each patient, labelled according to their hospitalisation status. (**D**). Protein-protein interaction network generated from stringDB demonstrating the relationships between LGALS9, LAMP3, PRSS8 and AGRN. (**E**). Pathway analysis plot showing the top 10 differentially expressed signalling pathways in hospitalised COVID-19 patients compared to non-hospitalised COVID-19 patients. Fold enrichment was measured on the *x*-axis, GO terms were listed on the *y*-axis, and the size and colour of the data points for each term were dependent on their −log10(*p*-value).

**Figure 3 biomolecules-14-01163-f003:**
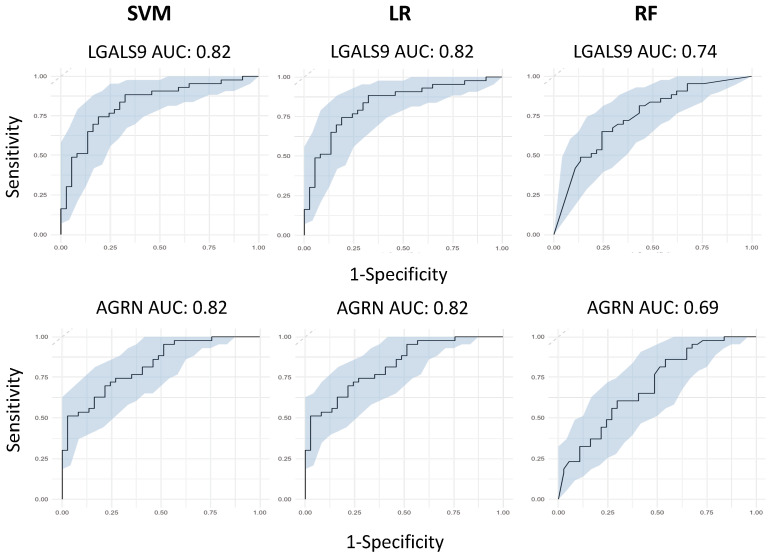
Univariate machine learning predictions for hospitalisation risk. Univariate ROC curves from SVM, LR and RF models for LGALS9, AGRN, PRSS8 and LAMP3 with labelled AUC scores and 95% confidence intervals shaded on the plot by bootstrap sampling the sensitivities and specificities 500 times.

**Figure 4 biomolecules-14-01163-f004:**
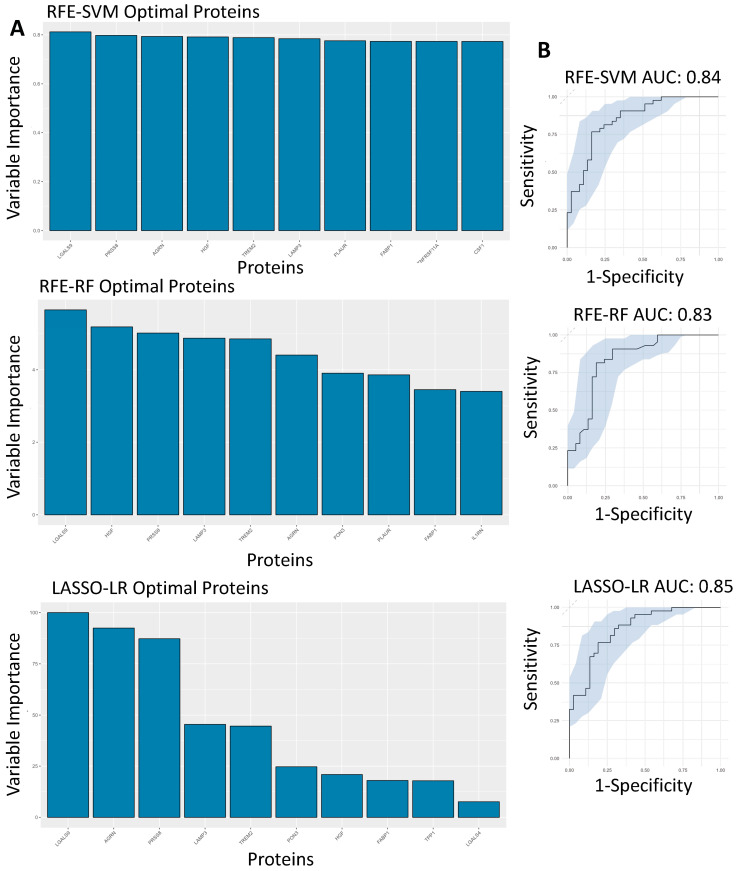
Feature-selected machine learning predictions for hospitalisation risk. (**A**). Variable importance plots of the RFE-SVM, RFE-RF and LASSO-LR models, ranking the top 10 most important features within the model according to their importance score. (**B**). Feature-selected ROC curves from RFE-SVM, RFE-RF and LASSO-LR using the optimal features, labelled AUC scores and 95% confidence intervals shaded on the plot by bootstrap sampling the sensitivities and specificities 500 times. (**C**). Confusion matrices generated for the RFE-SVM, RFE-RF and LASSO-LR models by comparing their actual predictions of hospitalised and non-hospitalised patients on unseen data not used for model training.

**Figure 5 biomolecules-14-01163-f005:**
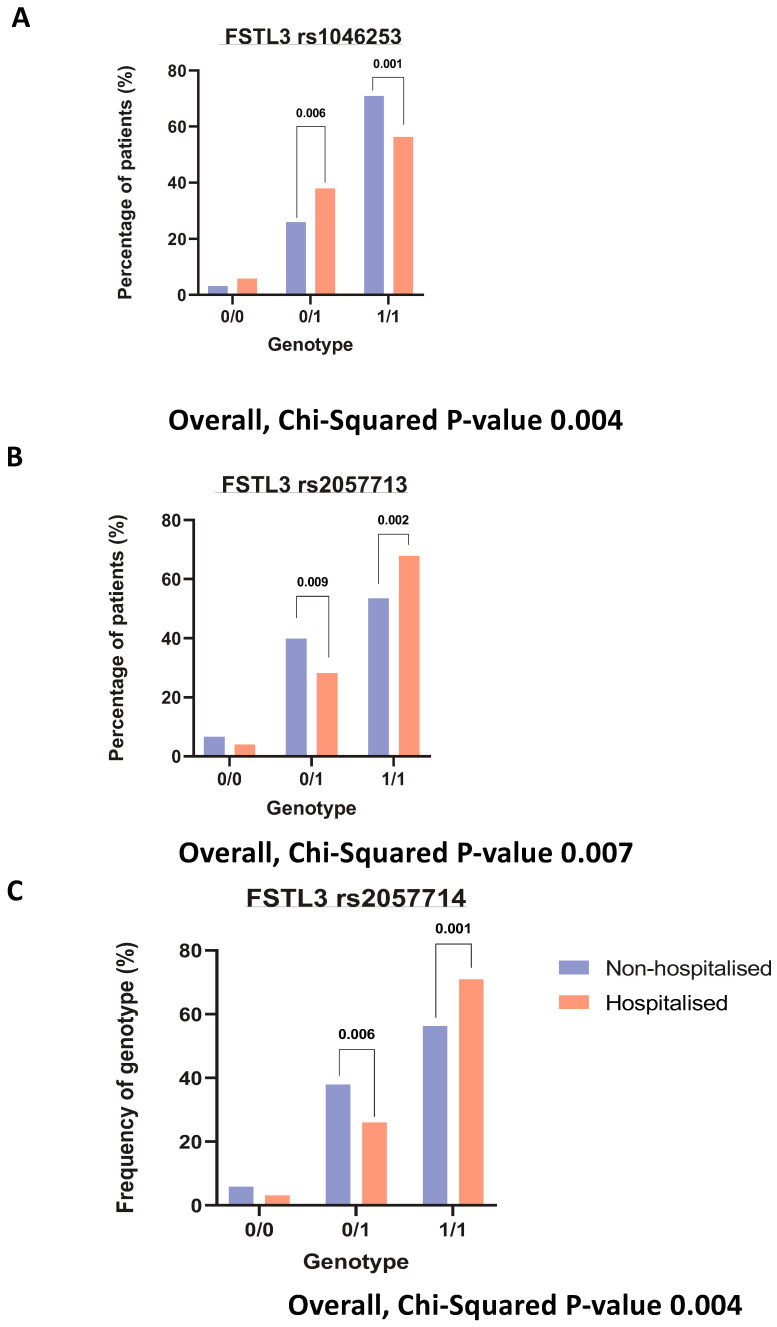
Genotyping analysis on key signatures. Bar charts demonstrating the percentage of each patient (hospitalised vs. non-hospitalised) and their genotypes, respective to each rsID. Where 0/0 represents the homozygous reference genotype, 0/1 represents the heterozygous genotype and 1/1 represents the homozygous alternative genotype. (**A**). *FSTL3* rs1046253 (**B**). *FSTL3* rs2057713. (**C**). *FSTL3* rs2057714.

**Table 1 biomolecules-14-01163-t001:** Base demographics for proteomic analysis. Base statistics for n = 400 COVID-19 patients who had their plasma proteins quantified. Table provides labelling for Age, Gender, and Severity across hospitalisation status.

	levels	Hospitalised	Non-Hospitalised	*p*-Value
**Age**	Mean (SD)	57.3 _(13.1)_	45 _(14.5)_	<0.001
**Severity**	Severe	214 _(100%)_	12 _(6.5%)_	<0.001
	Mild		174 _(93.5%)_	
	(%)			
**Gender**	Female	90 _(42.1%)_	109 _(58.6%)_	<0.004
	Male	120 _(56.1%)_	75 _(40.3%)_	
	Other	4 _(1.9%)_	2 _(1.1%)_	
	(%)			

**Table 2 biomolecules-14-01163-t002:** Statistics for top 10 differentially expressed proteins. Statistical data retrieved from testing NPX protein regulation between hospitalised COVID-19 patients and non-hospitalised COVID-19 patients, with the values for the top 10 most significant proteins identified.

Proteins	Log2FC	*p*-Value
LGALS9	0.686	4.537 × 10^−30^
PRSS8	0.724	8.208 × 10^−39^
AGRN	0.509	4.820 × 10^−29^
LAMP3	0.899	4.273 × 10^−27^
PLAUR	0.483	1.438 × 10^−24^
TREM2	0.855	1.096 × 10^−23^
TNFRSF11A	0.629	4.978 × 10^−23^
LAIR1	0.570	2.663 × 10^−22^
FSTL3	0.527	1.482 × 10^−21^
FABP1	1.166	1.648 × 10^−21^

## Data Availability

The raw data supporting the conclusions of this article will be made available by the authors on request.

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
