# Peer review of "Potential Plasma Proteins (LGALS9, LAMP3, PRSS8 and AGRN) as Predictors of Hospitalisation Risk in COVID-19 Patients"

_biomolecules, 2024, doi:10.3390/biom14091163_

Round 1

Reviewer 1 Report

Comments and Suggestions for Authors

This manuscript, entitled “LGALS9, LAMP3, PRSS8 and AGRN Predict Hospitalisation Risk in COVID-19 Patients”, aimed to reveal several plasma proteins as potential predictors for hospitalization risks of COVID-19 patients, with their predictive capabilities evaluated using three different machine learning models. Additionally,  possible patient genome SNPs related to the differentially expressed protein signatures were assessed. Overall, this manuscript is well-structured and well-written. Some details need to be clarified, and the title is suggested to be revised. Also, the figure resolution must be improved. Please see my line-to-line comments below.

Title: I suggest revising the title to “Potential plasma proteins as predictors of ….” This will make it more readable.

Author list: Please confirm and make sure the author list is complete, i.e., no corresponding author was included.

Line 75: Here it says the total number of patients recruited = 500, but in Methods the number = 519. Please clarify.

Line 81: Here the patient’s DNA was from saliva samples, but in Methods it said: “DNA from whole blood” (line 113). Please clarify. Also, the purpose of SNP identification was not clarified.

Line 119: No QC results was included for WGS.

Line 126: I believe no graphical abstract was submitted with this manuscript.

Line 140: Why is Bgzip included here?

Line 211: Please specify the column of P-value in Table 1.

Line 215: Please simplify Table 2 with significant digits. Table 2 may be suitable as a supplemental material.

Line 343: This sentence is not clear. Please clarify “SNPs across our identified signature”. Does it mean SNPs related to the 10 differentially expressed proteins?

Author Response

This manuscript, entitled “LGALS9, LAMP3, PRSS8 and AGRN Predict Hospitalisation Risk in COVID-19 Patients”, aimed to reveal several plasma proteins as potential predictors for hospitalization risks of COVID-19 patients, with their predictive capabilities evaluated using three different machine learning models. Additionally, possible patient genome SNPs related to the differentially expressed protein signatures were assessed.

Overall, this manuscript is well-structured and well-written. Some details need to be clarified, and the title is suggested to be revised.

Also, the figure resolution must be improved. Please see my line-to-line comments below.

Title: I suggest revising the title to “Potential plasma proteins as predictors of ….” This will make it more readable.

Response:

We acknowledge and agree with the reviewer and have both improved figure quality and revised the title as suggested.

Author list: Please confirm and make sure the author list is complete, i.e., no corresponding author was included.

Response:

We apologise and have added the corresponding author on MS file

Line 75: Here it says the total number of patients recruited = 500, but in Methods the number = 519. Please clarify.

Response:

Over recruitment was done to allow for dropouts. Total recruited was 519 but our ethics only allowed study on 500 patients. We have now corrected this in the methods section and also section 2.2.1

Line 81: Here the patient’s DNA was from saliva samples, but in Methods it said: “DNA from whole blood” (line 113). Please clarify. Also, the purpose of SNP identification was not clarified.

Response:

We thank the reviewer for pointing this out. This was a mistake on our part and is now corrected.  The purpose of SNPs identification was to gain insights into mechanistic changes driving severity of COVID-19. We added this clarification to the results section.

Line 119: No QC results was included for WGS.

Response: We did not add figures as this work was outsourced to Genuity Science. We can reassure the reviewer that all QC was performed in line with industry gold standards. Alignment quality was assessed, reads sorted, duplicates marked, indels realigned and base quality scores recalibrated. Sequences were uploaded onto the European Genome-phenome Archive (EGA).

Line 126: I believe no graphical abstract was submitted with this manuscript.

Response: We apologise to the reviewer for any confusion as the graphical abstract was previously denoted as “Study Design Overview”. We have since revised the title to “Graphical Abstract”.

Line 140: Why is Bgzip included here?

Response: Bgzip was included here in the methodology section as it was used to compress the overly large VCF files and format them as such to enable further analysis with bcftools. We apologise to the reviewer as the wording can be confusion and have revised the paragraph in question.

Line 211: Please specify the column of P-value in Table 1.

Response: We have revised the in-text referrals of both Tables 1 and 2, which now include specific column numbers that contain the appropriate statistics.

Line 215: Please simplify Table 2 with significant digits. Table 2 may be suitable as a supplemental material.

Response: We thank the reviewer for this suggestion and have since revised Table 2. Table 2 is now simplified and contains only the necessary relevant data of Protein Names, log2FC and p-values.

Line 343: This sentence is not clear. Please clarify “SNPs across our identified signature”. Does it mean SNPs related to the 10 differentially expressed proteins?

Response: We would like to clarify that the statement: Finally, querying several hundred SNPs across our identified signatures, is indeed the top 10 differentially expressed proteins being queried individually for several hundred SNPs each.

Reviewer 2 Report

Comments and Suggestions for Authors

This prognostic and biomarker study is very interesting and well-written. I have the following concerns that addressing them can improve the paper:

1. Fonts on many figures are very small and it is not possible to read them.

2. In machine learning section (methods), I expect authors elaborate more on the methods they have used.

3. Tables need to be improved visually.

Author Response

This prognostic and biomarker study is very interesting and well-written. I have the following concerns that addressing them can improve the paper:

Fonts on many figures are very small and it is not possible to read them.

Response: We thank the reviewer for this feedback and have increased the font size across all figures to make them more legible.

In machine learning section (methods), I expect authors elaborate more on the methods they have used.

Response: We thank the reviewer for this comment and have since added in additional information regarding each individual models tuning, family or kernel type. The section in its entirety now describes test train split, cross validation, hyper-tuning, family/kernel specifications, predictions on unseen data, confusion matrices generation and ROC curves.

Tables need to be improved visually.

Response: We appreciate the comments made by the reviewer regarding the layout of the tables. We have revised them as such and have kept the colours black and white to improve the visual aspects.

Reviewer 3 Report

Comments and Suggestions for Authors

Overall, this is effective study to predict hospitalization risk in COVID-19 using machine learning method. Based on Figure and description, I would recommend it for major revision. Here are some comments.

1. Author mentioned that LGALS9, LAMP3, AGRN and PRSS8 were found to be the most significant based on Figure 1B, however the quality of figure is very poor and hard to follow.

2. There is no clear explanation for Table 2. Why are some values negative for mean NPX Non-Hospitalized in the case of proteins PRSS8, LAMP3, etc.?

3. Except Figure 5, all Figures need serious editing. The resolution of the Figure is very bad, and the labels are not readable.

4.  It would be helpful and improve the manuscript if the author provided a workflow (describing the protocols) for the identification of significance proteins involved in predicting hospitalization Risk in COVID-19 patients.  

Author Response

Overall, this is effective study to predict hospitalization risk in COVID-19 using machine learning method. Based on Figure and description, I would recommend it for major revision. Here are some comments.

Author mentioned that LGALS9, LAMP3, AGRN and PRSS8 were found to be the most significant based on Figure 1B, however the quality of figure is very poor and hard to follow.

Response: We thank the reviewer for their comments about Figure 1B. We understand that unless the image is enlarged it can be hard to make out despite having a DPI of 350. To accommodate this, we have regenerated the figure and enlarged the text for more legibility.

There is no clear explanation for Table 2. Why are some values negative for mean NPX Non-Hospitalized in the case of proteins PRSS8, LAMP3, etc.?

Response: The data portrayed in Table 2 uses NPX or Normalised Protein Expression. This means all the values are scaled collectively and range between positive and negative intervals. We have since simplified Table 2 to contain only Protein Names, log2FC and p-values as suggested by the reviewer.

Except Figure 5, all Figures need serious editing. The resolution of the Figure is very bad, and the labels are not readable.

Response: We thank the reviewer for their feedback and have increased the font sizes across all figures to increase legibility. We would like to reassure the reviewer that the resolution on majority of the figures are of a DPI above 350

It would be helpful and improve the manuscript if the author provided a workflow (describing the protocols) for the identification of significance proteins involved in predicting hospitalization Risk in COVID-19 patients.  

Response: We thank the reviewer for this comment regarding a workflow based on our protocols. We have provided in our submission a graphical abstract that highlights the key protocols individually in a pipeline format, and have elaborated each one in the methodology segment of this manuscript.

Round 2

Reviewer 3 Report

Comments and Suggestions for Authors

I don't have further comments. It is now applicable for the publication.